# LATENT-SPACE REINFORCEMENT LEARNING FOR IMAGE SEGMENTATION

## ABSTRACT

Policy-gradient reinforcement learning is a theoretically grounded and empirically effective algorithm for boosting the performance of LLMs and MLLMs, while its adaptation to conventional vision tasks such as dense prediction remains marginal. In response, this work introduces a latent-space reinforcement learning framework designed for image segmentation with task-specific model architectures, aiming to investigate whether the advantages conferred by reinforcement learning in LLMs and MLLMs, including improved predictive performance, mitigation of forgetting and enhanced generalization, can be effectively transferred to conventional dense prediction tasks. The designed framework is instantiated with a latent-space policy network for feature representation modulation, a stabilized advantage formulation that underpins reliable policy updates, a segmentation-aligned reward formulation that quantifies segmentation quality, and a hybrid loss to enhance training stability and learning efficiency. The effectiveness of our proposed framework is validated through integration with widely used semantic segmentation models and empirical evaluation under cross-domain and continual learning settings. Across diverse and challenging benchmarks, the proposed framework delivers consistent performance gains, demonstrating its practical efficacy and highlighting its potential for broader application in future research.

## 1 INTRODUCTION

Policy-gradient reinforcement learning emerges as a powerful optimization paradigm that augments the expressiveness and generalization capacity of LLMs and MLLMs Ouyang et al. (2022); Liu et al. (2024); Yang et al. (2025). Besides, it enables sophisticated reasoning capabilities Hou et al. (2025); Yue et al. (2025a), improves factual reliability Roit et al. (2023); Tian et al. (2023); Jiao et al. (2025), and enforces alignment with user intent, safety considerations, and ethical principles, to name a few Achiam et al. (2017); Yuan et al. (2023); Dai et al. (2023); Tennant et al. (2024).

In contrast to its demonstrated impact on LLMs and their multimodal counterparts, the application of policy-gradient reinforcement learning to conventional vision domains, such as image segmentation, remains underexplored. To bridge this gap, this study aims to investigate the applicability of policy-gradient reinforcement learning to semantic segmentation and evaluates whether this integration can yield measurable benefits in predictive performance and generalization. Specifically, translating this paradigm into the domain of dense visual prediction introduces several unique challenges,

- **Action Space Complexity.** In LLMs and MLLMs, policy-gradient reinforcement learning typically operates over the token level, where each action corresponds to selecting a discrete token from a finite vocabulary Achiam et al. (2023); Yang et al. (2025); Zhou et al. (2025). By comparison, semantic segmentation requires the simultaneous prediction of dense, per-pixel labels over high-resolution input images, which gives rise to a high-dimensional and spatially correlated action space and thus significantly complicates policy optimization.

- **Pre-training Disparity.** Reinforcement learning in LLMs and MLLMs is typically applied after large-scale supervised or self-supervised pre-training on trillions of tokens, generating models that assign high probabilities to semantically meaningful token sequences and thus provide well-initialized action distributions for subsequent policy optimization Dubey et al. (2024); Liu et al. (2024); Yang et al. (2025). In contrast, segmentation models are typically

Figure 1: Illustration of latent-space reinforcement learning in semantic segmentation.

initialized from vision backbones pre-trained on either image-level classification tasks, *e.g.*, ResNet and ViT Long et al. (2015); Zheng et al. (2021), or self-supervised objectives such as masked image modeling He et al. (2022); Bao et al. (2021), over comparatively smaller-scale image datasets like ImageNet Deng et al. (2009). The mismatch between pre-training (image-level understanding) and fine-tuning (pixel-level decision-making), combined with the limited scale of vision pre-training data, results in poorly initialized action distributions for dense prediction tasks. This significantly impairs the stability of reinforcement learning and renders early-stage policy optimization particularly susceptible to divergence.

- **Reward Granularity and Sparsity.** Unlike LLMs and MLLMs, where rewards align with token-level generation, segmentation metrics like mIoU and Dice provide only image-level feedback. This coarse supervision hampers credit assignment across pixel-wise actions and leads to unstable policy updates.

Building upon the challenges outlined above, this work designs a latent-space reinforcement learning framework tailored for image segmentation. Operating in the latent space offers two key advantages: first, the spatial resolution of feature maps is considerably lower than that of the input image, which reduces the size of the action space and alleviates optimization complexity. Moreover, feature-level actions exhibit greater flexibility, as variations in feature representations can correspond to the same semantic class, whereas pixel-level actions are strictly tied to fixed label assignments.

Figure 1 illustrates the overall pipeline of the proposed algorithm. Given a segmentor comprising an encoder and decoder, we introduce a policy module that observes the latent-space features produced by the encoder and generates modulation signals to guide features adaptation prior to decoding. The segmentation outputs are then adopted to compute reward signals, which in turn drive policy updates via standard policy-gradient optimization Sutton et al. (1999). To further address the aforementioned challenges, we introduce several targeted design components within the proposed pipeline, including a task-aligned reward function to estimate prediction quality, a stabilized advantage formulation with temporal smoothing to reduce reward variance and ensure stable and effective policy updates, and a hybrid loss that combines policy-gradient objectives and supervised segmentation signals, similar in spirit to VAPO Yue et al. (2025b), to mitigate pre-training disparity and reward sparsity issues.

The principal contributions of this work can be articulated as follows,

- We present the first latent-space formulation of policy-gradient reinforcement learning for image segmentation, indicating that key benefits observed in LLMs and MLLMs, including improved performance, reduced forgetting, and enhanced generalization, can be effectively extended to conventional segmentation settings.
- To address the unique issues of applying reinforcement learning to dense visual prediction, we introduce a latent-space reinforcement learning framework comprising a policy network for modulating intermediate feature representations, a stabilized advantage formulation for robust policy updates, a task-aligned reward function for prediction quality estimation, and a hybrid loss that combines policy-gradient and supervised signals to alleviate pre-training mismatch and reward sparsity.
- We empirically validate the proposed algorithm across multiple segmentation architectures and standard benchmarks, indicate its effectiveness in continual learning and cross-domain settings, and conduct ablations to evaluate the role of each design choice.

We hope this study will inspire future research on integrating policy-gradient reinforcement learning into conventional vision tasks beyond semantic segmentation.

## 2 RELATED WORK

**Semantic Segmentation.** Semantic segmentation is the process of assigning a label to each pixel in a given image so that pixels with the same label share certain visual characteristics or are associated with the same semantic category. In the deep learning era, semantic segmentation is usually tackled through encoder-decoder architectures Long et al. (2015). Within this framework, a substantial body of influential studies continues to emerge and could be categorized according to their methodological focuses, including contextual information aggregation Zhao et al. (2017); Chen et al. (2017); Yuan et al. (2020); Jin et al. (2022; 2023), objective function optimization Seo et al. (2020); Jadon (2020); Zhao et al. (2020); Eelbode et al. (2020); Shirokikh et al. (2020), feature encoder enhancement Liu et al. (2021); Dosovitskiy et al. (2020); He et al. (2022); Xia et al. (2024), and segmentation decode flow reconfiguration Cheng et al. (2022); Zhang et al. (2021); Zhou et al. (2022).

In addition to methodological advances, semantic segmentation research continues to expand across diverse evaluation settings, *e.g.*, cross-domain semantic segmentation which focuses on performance under domain shifts between training and testing data Lv et al. (2020); Gong et al. (2023); Luo et al. (2024), continual semantic segmentation which aims to incrementally learn new classes or domains without catastrophic forgetting Douillard et al. (2021); Toldo et al. (2024); Zhang et al. (2024); Yin et al. (2025), few-shot and zero-shot segmentation which explore generalization to new classes with limited or no labeled examples Wang et al. (2019); Bucher et al. (2019); Ding et al. (2022); He et al. (2024), and semi-supervised or weakly supervised segmentation which reduce dependence on dense annotations by leveraging unlabeled or weakly labeled data Wei et al. (2016); Ouali et al. (2020).

Building upon established network architectures and diverse evaluation settings, this study explores the integration of policy-gradient reinforcement learning as a means to enhance model performance.

**Reinforcement Learning.** Reinforcement learning (RL) is a learning framework in which an agent learns to make sequential decisions by interacting with the environment and receiving reward-based feedback Kaelbling et al. (1996). Based on their underlying learning formulation, RL algorithms are typically categorized into three major classes: *value-based methods*, which estimate value functions to guide action selection Hester et al. (2018); Hou et al. (2017); Sun et al. (2022); Lobel et al. (2023), *policy-based methods*, which directly optimize a parameterized policy Schulman et al. (2015); Sutton et al. (1999); Schulman et al. (2017); Shao et al. (2024); Yu et al. (2025), *actor-critic paradigms*, which combine value estimation and policy learning to enable more stable and efficient optimization Grondman et al. (2012); Andrychowicz et al. (2021); Duan et al. (2021); Zanette et al. (2021); Ma et al. (2025). This study investigates the application of policy-based methods to conventional vision tasks, with a particular focus on semantic segmentation.

**RL for Semantic Segmentation.** Early efforts in this area adopt RL to frame semantic segmentation as a sequential decision-making process Casanova et al. (2020); Duan et al. (2022); Tian et al. (2022). For instance, RL-CoSeg Duan et al. (2022) formulates image co-segmentation as a Markov Decision Process and leverages an asynchronous advantage actor-critic strategy to iteratively optimize region boundaries across related images. Recent studies continue to expand RL applications to a variety of segmentation scenarios, including medical imaging Liu et al. (2025a), robotics Zhang et al. (2025), and reasoning-aware segmentation Liu et al. (2025b). Among prior efforts, PixelDRL-MG Liu et al. (2025a) proposes a pixel-level asynchronous actor-critic framework wherein each pixel is treated as an agent, and a shared policy network progressively refines outputs from coarse to fine.

In contrast to prior approaches, we investigate the integration of policy-gradient RL directly within conventional segmentation networks without relying on large foundation models. To the best of our knowledge, this is the first work to incorporate RL-based policy optimization into the latent space of semantic segmentation networks.

## 3 METHODOLOGY

In this section, we begin by presenting the semantic segmentation paradigm, which is built upon an encoder-decoder architecture forming the backbone of the proposed approach. We then elaborate on the incorporation of policy-gradient RL into the latent feature space, emphasizing its contribution to enhancing representational capacity and segmentation accuracy. At last, we describe the key designs of our framework, *e.g.*, the advantage formulation, reward design and hybrid loss.

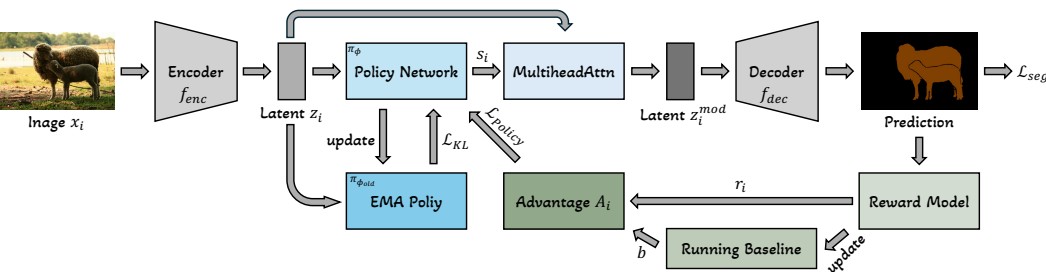

Figure 2: Overview of the introduced latent-space RL framework for semantic segmentation. Latent-space features $z_i$ extracted by the encoder are modulated by a policy network $\pi_\phi$ through stochastic Gaussian sampling, and decoded into segmentation outputs. Rewards $r_i$ and advantages $A_i$ regulate policy optimization via a clipped objective with a KL penalty against an EMA-stabilized reference.

### 3.1 SEMANTIC SEGMENTATION PIPELINE

We begin by formalizing the task of semantic segmentation. Let $\mathcal{X} \subset \mathbb{R}^{H \times W \times 3}$ denote the space of input images and let $\mathcal{Y} \subset \{1, \ldots, K\}^{H \times W}$ represent the space of pixel-wise segmentation masks in which $K$ is the number of semantic categories. The goal is to learn a composite mapping,

$$f_{\text{dec}} \circ f_{\text{enc}} : \mathcal{X} \to \mathcal{Y}, \tag{1}$$

where:

- $f_{\text{enc}} : \mathcal{X} \to \mathcal{Z}$ is the **encoder**, mapping input images to latent-space features $\mathcal{Z} \subset \mathbb{R}^{h \times w \times d}$, where $h$ and $w$ denote spatial dimensions and $d$ the feature channels.
- $f_{\text{dec}} : \mathcal{Z} \to \mathcal{Y}$ is the **decoder**, mapping latent features to pixel-wise segmentation masks.

Given a dataset $(x_i, y_i)_{i=1}^N$ consisting of input images $x_i$ and their corresponding ground truth masks $y_i$, the segmentor is trained by minimizing a pixel-wise loss, typically the cross-entropy loss,

$$\mathcal{L}_{\text{seg}}(\theta) = -\frac{1}{N} \sum_{i=1}^N \sum_{p \in \Omega} \sum_{k=1}^K \mathbf{1}(y_i^p = k) \log P(f_{\text{dec}}(f_{\text{enc}}(x_i))^p = k), \tag{2}$$

where $\Omega$ denotes the set of pixel locations, $y_i^p$ is the ground-truth label at location $p$ and the function $\mathbf{1}(y_i^p = k)$ is an indicator that equals $1$ if the ground-truth label at location $p$ belongs to class $k$, and $0$ otherwise. $f_{\text{dec}}(f_{\text{enc}}(x_i))^p$ is the predicted probability of class $k$ at location $p$. The parameter set $\theta$ includes all learnable weights in $f_{\text{enc}}$ and $f_{\text{dec}}$.

The intermediate representation $z_i = f_{\text{enc}}(x_i) \in \mathbb{R}^{h \times w \times d}$ encodes semantic information of the input image and defines the latent space. We next present the proposed RL algorithm that operates directly on this latent space $z_i \in \mathbb{R}^{h \times w \times d}$.

### 3.2 LATENT-SPACE REINFORCEMENT LEARNING

To move beyond canonical supervised learning formulations in semantic segmentation, we propose a latent-space RL framework that adaptively modulates intermediate feature representations through policy optimization, after which the modulated features are decoded into segmentation predictions.

**MDP Formulation.** We recast this problem in the language of Markov Decision Processes (MDPs). Unlike autoregressive language models Achiam et al. (2023); Liu et al. (2024), wherein RL operates over sequential token generation with long trajectories, semantic segmentation produces predictions simultaneously across all pixels of $x_i$. This eliminates temporal dependencies and renders semantic segmentation more naturally aligned with a *contextual bandit*, which can be viewed as a degenerate one-step MDP. Formally, the MDP is defined as $(\mathcal{S}, \mathcal{A}, \mathcal{R}, \mathcal{P}, \gamma)$, where:

- the state $\mathcal{S}$ is the latent-space feature representations $z_i = f_{\text{enc}}(x_i)$,
- the action $\mathcal{A}$ is the modulation signal $s_i$ sampled from a Gaussian policy $\pi_\phi$,

- the transition $\mathcal{P}$ is the deterministic decoding $f_{\text{dec}}(z_i, s_i)$,
- the reward $\mathcal{R}$ is computed from task metrics, *e.g.*, mIoU and Dice,
- the discount factor is $\gamma = 1$, as each image forms a one-step decision.

This contextual-bandit view treats each image as an independent episode where the policy observes latent-space features, selects modulation actions, and receives immediate rewards.

**Overview of the Latent-space RL Framework.** As illustrated in Figure 2, let $z_i$ denote the latent-space representation extracted from an input image $x_i$ using the encoder. Since $z_i$ encodes high-level semantic patterns, it serves as the input to a latent-space policy network $\pi_\phi$. This policy network $\pi_\phi$ treats each feature channel as a Gaussian distribution and provides a stochastic modulation signal $s_i$ by sampling from $\mathcal{N}(\mu_i, \sigma_i^2)$, where $\mu_i$ and $\sigma_i^2$ are predicted from a compressed projection of $z_i$. To enable differentiable sampling, the reparameterization trick is applied. The sampled signal $s_i$ is then adopted to modulate $z_i$ through a lightweight cross-attention mechanism, producing policy-adaptive features for decoding into segmentation predictions.

To optimize $\pi_\phi$, we leverage Proximal Policy Optimization (PPO) Schulman et al. (2017), replacing the standard advantage estimator with our introduced stabilized advantage formulation. Specifically, after computing the segmentation logits from the decoder $f_{\text{dec}}$, prediction quality is assessed utilizing task-specific metrics, *e.g.*, mIoU and Dice. These metrics are transformed into scalar rewards $r_i$ for $x_i$. To stabilize updates of $\pi_\phi$, a running baseline $b$ is maintained, and the advantage is computed as,

$$A_i = \text{clamp}(r_i - b, a_{\min}, a_{\max}), \tag{3}$$

where $a_{\min}$ and $a_{\max}$ are thresholds introduced to control gradient variance. The policy objective is defined as,

$$\mathcal{L}_{\text{Policy}} = -\mathbb{E}\left[\min\left(\rho_i A_i, \text{clip}(\rho_i, 1 - \epsilon, 1 + \epsilon)A_i\right)\right], \tag{4}$$

with $\rho_i = \exp(\log \pi_\phi(s_i) - \log \pi_{\phi_{\text{old}}}(s_i))$ representing the importance ratio between the current and EMA-stabilized reference policies. To constrain policy shift, a KL divergence penalty is imposed,

$$\mathcal{L}_{\text{KL}} = \beta \cdot \text{KL}(\pi_{\phi_{\text{old}}} \| \pi_\phi). \tag{5}$$

The overall training objective integrates supervised segmentation with RL-based regularization,

$$\mathcal{L}_{\text{total}} = \mathcal{L}_{\text{seg}} + \lambda_{\text{Policy}} \mathcal{L}_{\text{Policy}} + \lambda_{\text{KL}} \mathcal{L}_{\text{KL}}, \tag{6}$$

where $\mathcal{L}_{\text{seg}}$ is the standard pixel-wise loss and $\lambda_{\text{Policy}}, \lambda_{\text{KL}}$ are balancing coefficients.

Table 1: Comparison of hybrid training regimes for integrating reinforcement learning into semantic segmentation. Experiments are conducted using FCN on PASCAL VOC 2012. Results are reported as mean $\pm$ std of mIoU (%) over five runs.

| Optimization Schedule | Mean $\pm$ Std |
|---|---|
| Pre-training on ImageNet $\rightarrow$ SFT on PASCAL VOC 2012 $\rightarrow$ RL on PASCAL VOC 2012 | $68.5 \pm 0.90$ |
| Pre-training on ImageNet $\rightarrow$ only RL on PASCAL VOC 2012 | $6.4 \pm 3.33$ |
| Pre-training on ImageNet $\rightarrow$ joint SFT + RL on PASCAL VOC 2012 | $\mathbf{78.5 \pm 0.43}$ |

**Discussion of Hybrid Objectives.** Eq. (6) facilitates RL to directly regulate latent-space modulation through task-level rewards, while maintaining the efficiency of supervised feature learning.

In the literature on LLMs and MLLMs, hybrid training typically follows one of three paradigms,

- pre-training followed by supervised fine-tuning (SFT) and then RL Achiam et al. (2023),
- pre-training followed directly by RL Liu et al. (2024),
- pre-training followed by alternating or joint SFT and RL Dey et al. (2021) as formulated in Eq. 6 (*i.e.*, pre-train on ImageNet and then joint SFT and RL on task-specific benchmarks).

We assess the applicability of these hybrid optimization paradigms to semantic segmentation through a series of preliminary experiments. As shown in Table 1, the joint SFT and RL strategy, instantiated in our designed formulation Eq. 6, obtains the highest performance, exhibiting both effectiveness and stability. In contrast, directly applying RL without prior SFT induces pronounced task misalignment, as the pre-trained model lacks sufficient adaptation to the segmentation objective. This misalignment

results in a suboptimal initialization of the RL action space, culminating in severely degraded model performance. Besides, applying RL subsequent to SFT obtains only modest performance, likely due to over-fitting introduced during SFT, a phenomenon exacerbated by the limited size of segmentation datasets compared to those exploited in LLMs or MLLMs training. Once over-fitting occurs, RL has limited capacity to further improve the model.

In our preliminary experiments, similar observations hold when adopting backbones pre-trained with self-supervised objectives including masked image modeling He et al. (2022). This can be attributed to the misalignment between pre-training and semantic segmentation objectives, the smaller size of vision pre-training datasets and the random initialization of the decoder, which collectively limit the quality of action initialization for RL.

**Latent-Space Policy Network.** We design a latent-space policy network $\pi_\phi$ to generate a stochastic modulation signal $s_i$ conditioned on latent-space features $z_i$. The core idea is to model each feature channel, which typically encodes a distinct semantic pattern, as a Gaussian distribution, from which a channel-specific modulation signal is sampled to guide decoding process adaptively. By modeling each channel as an independent Gaussian, the policy can selectively refine the semantic information of $z_i$ such as object parts or textures prior to decoding.

To achieve this, $\pi_\phi$ first applies a sequence of $L$ convolutional blocks to $z_i$,

$$\text{ConvProj}_\phi(z_i) = \mathcal{B}_\phi^{(L)} \circ \cdots \circ \mathcal{B}_\phi^{(2)} \circ \mathcal{B}_\phi^{(1)}(z_i), \tag{7}$$

followed by adaptive average pooling to a fixed spatial resolution, resulting in $z_i' \in \mathbb{R}^{P_h \times P_w \times d}$. The block $\mathcal{B}_\phi^{(\ell)}$ for $\ell = 1, \ldots, L$ is defined as,

$$\mathcal{B}_\phi^{(\ell)}(z_i) = \text{ReLU}(\text{BN}^{(\ell)}(\text{Conv}^{(\ell)}(z_i))), \tag{8}$$

where $\phi$ denotes the learnable parameters. In our implementation, we use $L = 4$. The pooled feature $z_i'$ is then flattened across spatial dimensions to obtain a channel-wise descriptor,

$$\overline{z}_i = \text{Flatten}(z_i') \in \mathbb{R}^{(P_h \cdot P_w) \times d}. \tag{9}$$

From this descriptor, $\pi_\phi$ predicts the mean and log standard deviation for each feature channel,

$$\mu_i = \text{fc}_\mu(\overline{z}_i), \quad \log \sigma_i = 4 \cdot \tanh(\text{fc}_{\log \sigma}(\overline{z}_i)), \tag{10}$$

where both $\mu_i, \log \sigma_i \in \mathbb{R}^d$, and the scaling factor 4 is adopted to stabilize exploration. A modulation signal is then sampled leveraging the reparameterization trick which enables differentiable stochastic sampling, thereby supporting effective exploration and facilitating gradient-based credit assignment from reward signals,

$$s_i = \mu_i + \sigma_i \cdot \epsilon, \quad \epsilon \sim \mathcal{N}(0, 1), \quad s_i \in \mathbb{R}^{h \times w \times d}. \tag{11}$$

The log-probability of the sampled modulation $s_i$ is given by,

$$\log \pi_\phi(s_i) = \sum_{h,w,d} \log \mathcal{N}(s_i^{(h,w,d)} \mid \mu_{i,d}, \sigma_{i,d}^2), \tag{12}$$

and is used for computing the policy gradient during training.

**Attention-Based Modulation.** To inject $s_i$ into the semantic feature space, we propose an attention-based modulation module that adaptively refines $z_i$ conditioned on $s_i$. We begin by aligning $s_i$ with $z_i$ through a lightweight convolutional projection, ensuring dimensional and semantic compatibility,

$$s_i' = \mathcal{B}(s_i) = \text{ReLU}(\text{BN}(\text{Conv}(s_i))), \ s_i' \in \mathbb{R}^{h \times w \times d}. \tag{13}$$

The transformed signal $s_i'$ is then used to modulate $z_i$ via a multi-head attention mechanism,

$$z_i^{\text{mod}} = \text{MultiheadAttention}(z_i - s_i', z_i, z_i), \tag{14}$$

where $z_i^{\text{mod}}$ is the resulting policy-adaptive representation that incorporates modulation signals into the semantic context. To improve training stability, we apply a learnable scaling factor $\alpha$ to $z_i^{\text{mod}}$ and concatenate it with $z_i$ before feeding into the decoder.

**Reward Design.** To effectively guide latent-space reinforcement learning in semantic segmentation, we adopt a composite reward function that integrates multiple task-relevant evaluation metrics.

Specifically, the reward comprises the following components,

- $r^{\text{IoU}}$: Measures region-level alignment adopting the Intersection-over-Union (IoU) between predicted and ground-truth masks.
- $r^{\text{Dice}}$: Focuses on foreground overlap, helping in class-imbalanced scenarios.
- $r^{\text{CE}}$: Inverse of cross-entropy loss, promoting pixel-wise accuracy.
- $r^{\text{Boundary}}$: Measures edge accuracy by comparing predicted and ground-truth boundaries.

The final reward signal used for policy optimization is defined as,

$$r = \lambda_{\text{IoU}} r^{\text{IoU}} + \lambda_{\text{Dice}} r^{\text{Dice}} + \lambda_{\text{CE}} r^{\text{CE}} + \lambda_{\text{Boundary}} r^{\text{B}}, \tag{15}$$

where $\lambda_{\text{IoU}}, \lambda_{\text{Dice}}, \lambda_{\text{CE}}, \lambda_{\text{Boundary}}$ are weighting coefficients that balance the influence of each reward term. In our implementation, we use $\lambda_{\text{IoU}} = \lambda_{\text{Dice}} = \lambda_{\text{CE}} = \lambda_{\text{Boundary}} = 1$.

**Advantage Formulation.** To stabilize training and reduce variance in policy gradient estimates, we utilize a momentum-based baseline to compute the advantage function, which quantifies the relative quality of the action of the current policy.

Given the scalar reward $r_i$ for each input $x_i$, we maintain an exponential moving average baseline $b$,

$$b \leftarrow \tau b + (1 - \tau) \cdot \bar{r}, \tag{16}$$

where $\tau \in [0, 1]$ is the smoothing coefficient and $\bar{r}$ is the mean reward across the current mini-batch. The advantage is then computed by subtracting the baseline,

$$A_i = r_i - b. \tag{17}$$

To further enhance training stability, we clip the advantage values within a bounded interval,

$$A_i \leftarrow \text{clamp}(A_i, \, a_{\min}, \, a_{\max}), \tag{18}$$

mitigating the influence of outlier gradients.

**Historical Policy.** To estimate the importance ratio $\rho_i$ in Eq. (4), and the KL regularization term in Eq. (5), we maintain a historical policy network $\pi_{\phi_{\text{old}}}$ as a stable reference. This network is updated using an exponential moving average (EMA) of the current policy parameters,

$$\phi_{\text{old}} \leftarrow \eta \cdot \phi_{\text{old}} + (1 - \eta) \cdot \phi, \tag{19}$$

where $\eta \in [0, 1]$ controls the update momentum.

This EMA update gives rise to a stable reference policy for computing $\rho_i$ and KL divergence, helping to reduce gradient variance and prevent policy collapse. Importantly, $\pi_{\phi_{\text{old}}}$ is used only for evaluation and excluded from gradient updates, ensuring temporal consistency during training.

## 4 EXPERIMENTS

This section presents a comprehensive experimental evaluation of the proposed algorithm. We begin by detailing the experimental setup such as benchmark datasets, baseline models and implementation specifics. We then report the primary results in comparison with competitive baselines, followed by systematic ablation studies to analyze the contribution of each design element in our framework.

### 4.1 EXPERIMENTAL SETUP

**Benchmark Datasets.** We evaluate on ADE20K Zhou et al. (2017), Cityscapes Cordts et al. (2016), and PASCAL VOC Long et al. (2015), three semantic segmentation benchmarks with varying scene types and annotation granularity. ADE20K includes 27,574 scene-centric images across 150 classes, Cityscapes provides 5,000 finely annotated urban street-scene samples, annotated with 30 semantic categories, commonly evaluated over a standard subset of 19 semantic labels. PASCAL VOC offers 21-class annotations, and we use its widely adopted augmented training set with 10,582 images.

**Baseline Models.** We compare our method with representative baselines under various settings. For supervised semantic segmentation, we adopt FCN Long et al. (2015), SETR Zheng et al. (2021) and Segformer Xie et al. (2021). In the continual semantic segmentation settings, we benchmark against

Table 2: Performance improvements of Latent-Space RL in supervised semantic segmentation tasks across diverse segmentation architectures. Parentheses indicate gains over baseline models.

| Method | Backbone | ADE20K (mIoU) | Cityscapes (mIoU) | PASCAL VOC (mIoU) |
|---|---|---|---|---|
| FCN | ResNet-50 | 37.0 | 75.2 | 67.8 |
| FCN + Latent-Space RL | ResNet-50 | 44.1 (**+7.1**) | 79.8 (**+4.6**) | 78.5 (**+10.7**) |
| Segformer | MiT-B1 | 42.3 | 78.6 | 77.5 |
| Segformer + Latent-Space RL | MiT-B1 | 43.4 (**+1.1**) | 79.8 (**+1.2**) | 78.7 (**+1.2**) |
| Segformer | MiT-B3 | 48.3 | 82.0 | 82.0 |
| Segformer + Latent-Space RL | MiT-B3 | 49.8 (**+1.5**) | 83.3 (**+1.3**) | 83.5 (**+1.5**) |
| SETR-Naive | ViT-Large | 48.4 | 78.4 | 84.5 |
| SETR-Naive + Latent-Space RL | ViT-Large | 49.5 (**+1.1**) | 80.2 (**+1.8**) | 85.5 (**+1.0**) |

Table 3: Performance improvements of Latent-Space RL in continual semantic segmentation across varying settings on PASCAL VOC 2012 and ADE20K. Parentheses indicate gains over baselines.

| Dataset | Method | 15-5 (2 steps) | | | 15-1 (6 steps) | | | 10-1 (11 steps) | | |
|---|---|---|---|---|---|---|---|---|---|---|
| | | *0-15* | *16-20* | *all* | *0-15* | *16-20* | *all* | *0-10* | *11-20* | *all* |
| VOC 2012 | PLOP | 76.2 | 49.6 | 69.9 | 66.9 | 19.7 | 55.7 | 46.4 | 15.3 | 31.6 |
| | PLOP + Latent-Space RL | 78.2 | 59.2 | 73.7 (**+3.8**) | 71.3 | 34.1 | 62.4 (**+6.7**) | 60.3 | 22.6 | 42.3 (**+10.7**) |

| Dataset | Method | 100-50 (2 steps) | | | 100-10 (6 steps) | | | 100-5 (11 steps) | | |
|---|---|---|---|---|---|---|---|---|---|---|
| | | *1-100* | *101-150* | *all* | *1-100* | *101-150* | *all* | *1-100* | *101-150* | *all* |
| ADE20K | PLOP | 41.9 | 14.9 | 32.9 | 40.5 | 13.6 | 31.6 | 39.1 | 7.8 | 28.8 |
| | PLOP + Latent-Space RL | 44.1 | 19.2 | 35.8 (**+2.9**) | 42.2 | 17.5 | 34.0 (**+2.4**) | 41.6 | 15.0 | 32.7 (**+3.9**) |

PLOP Douillard et al. (2021). For cross-domain evaluation, we use FCN as the base model to assess generalization across domains.

**Reproducibility.** Our reinforcement learning framework is implemented in PyTorch and trained on 2×NVIDIA H200 GPUs (141 GB memory per card). Inference is conducted on a single H200 GPU. We will release the complete source code to facilitate reproducibility.

**Evaluation Metrics.** Following standard practice, we utilize mean Intersection over Union (mIoU) as the main evaluation metric. For continual semantic segmentation, we report three mIoU variants, *i.e.*, mIoU on the initial class set $\mathcal{C}^0$, mIoU on the incremental class sets $\{\mathcal{C}^1, \ldots, \mathcal{C}^T\}$ and mIoU on all learned classes $\{\mathcal{C}^0, \ldots, \mathcal{C}^T\}$, following established protocols Douillard et al. (2021).

**Implementation Details.** For both supervised and cross-domain semantic segmentation, we adhere to the default training configurations provided by SSSegmentation Jin (2023), such as segmentation model initialization, optimization settings and data augmentation strategies. For continual semantic segmentation, we utilize the official implementation and protocols of PLOP Douillard et al. (2021). All experiments are repeated five times with different random seeds (*i.e.*, from zero to four), and the reported results correspond to the average performance across runs.

## 4.2 MAIN RESULTS

To validate the effectiveness of the proposed latent-space RL framework for semantic segmentation, we conduct extensive experiments under three settings, including supervised semantic segmentation, continual learning and cross-domain generalization, using standard benchmarks and architectures.

As shown in Table 2, integrating latent-space RL consistently improves performance across diverse architectures and datasets. For instance, FCN with latent-space RL achieves mIoU gains of 7.1% on ADE20K, 4.6% on Cityscapes and 10.7% on PASCAL VOC. These results suggest that RL in latent space enhances the representational capacity and segmentation quality of standard architectures.

Table 3 reports results under incremental learning protocols on PASCAL VOC and ADE20K. Latent-space RL yields substantial improvements over PLOP, particularly in longer sequences such as 10-1 and 100-5, where it reduces forgetting and preserves performance on earlier classes. This indicates that RL benefits observed in LLMs and MLLMs like improved memory retention can be effectively transferred to conventional vision tasks Rafailov et al. (2023); Dai et al. (2023).

In Table 4, we evaluate cross-domain performance leveraging models trained on Cityscapes and LIP, where LIP Liang et al. (2018) is a single-person human parsing benchmark comprising 50K images with 19 semantic human part categories, and assess their generalization to three target domains with

Table 4: mIoU improvements of Latent-Space RL in cross-domain semantic segmentation. Outputs are reported on Dark Zurich and Nighttime Driving using models trained on Cityscapes and on CIHP using models trained on LIP. Parentheses indicate mIoU gains over the FCN baseline.

| Method | Urban Scene Parsing (*Cityscapes train*) | | Human Parsing (*LIP train*) |
| | *Dark Zurich val* | *Nighttime Driving test* | *CIHP val* |
| --- | --- | --- | --- |
| FCN | 10.7 | 17.9 | 27.2 |
| FCN + Latent-Space RL | 15.9 (+5.2) | 26.1 (+8.2) | 29.0 (+1.8) |

Table 5: Ablation studies on (a) reward function and (b) baseline design for advantage estimation in our latent-space RL framework on PASCAL VOC.

| (a) Reward Function | | (b) Baseline Design | |
| Reward Configuration | PASCAL VOC (mIoU) | Baseline Formulation | PASCAL VOC (mIoU) |
| --- | --- | --- | --- |
| FCN | 67.8 | FCN | 67.8 |
| $r^{\text{IoU}}$ only | 77.1 | $b = 0$ | $76.1 \pm 0.95$ |
| $r^{\text{IoU}} + r^{\text{Dice}}$ | 77.5 | $b = \bar{r}$ | $77.0 \pm 0.78$ |
| $r^{\text{IoU}} + r^{\text{Dice}} + r^{\text{CE}}$ | 77.9 | $b \leftarrow \tau b + (1-\tau)\bar{r}$ | $\mathbf{78.5 \pm 0.43}$ |
| $r^{\text{IoU}} + r^{\text{Dice}} + r^{\text{CE}} + r^{\text{Boundary}}$ | **78.5** | | |

distribution shifts. Dark Zurich and Nighttime Driving Wood (2020) represent low-light urban scene benchmarks that differ markedly from the daytime settings of Cityscapes, introducing domain shifts in illumination and appearance. CIHP Gong et al. (2018) is a multi-human parsing dataset, deviating substantially from the single-human focus of LIP. Our latent-space RL framework obtains consistent mIoU improvements across all target domains, *i.e.*, 1.8% on CIHP, 5.2% on Dark Zurich, and 8.2% on Nighttime Driving, highlighting its capacity to enhance cross-domain generalization.

These results demonstrate that key advantages of policy-gradient RL in LLMs and MLLMs such as improved predictive accuracy, mitigation of catastrophic forgetting and enhanced generalization can be effectively transferred to conventional vision tasks like semantic segmentation.

### 4.3 ABLATION STUDIES

**Reward Configuration.** Table 5 presents an ablation study examining the impact of different reward components in our latent-space RL framework. Beginning with the IoU-based reward term $r^{\text{IoU}}$, we progressively incorporate additional task-relevant rewards, *i.e.*, $r^{\text{Dice}}$, $r^{\text{CE}}$, and $r^{\text{Boundary}}$. Each added component contributes complementary supervision, leading to consistent improvements in mIoU.

**Baseline Design.** We investigate the impact of different baseline designs on the stability and efficacy of policy optimization within our framework. As summarized in Table 5, omitting a baseline ($b = 0$) yields suboptimal performance, as all rewards are treated as positive signals, leading to uncalibrated and potentially overconfident policy updates. Introducing the mini-batch mean baseline (*i.e.*, $b = \bar{r}$) partially mitigates this problem by normalizing advantage estimates within each batch. Nevertheless, given the typically small batch sizes adopted in semantic segmentation and the high variance in per-image prediction quality, this approach introduces considerable estimation noise, ultimately limiting its effectiveness. In contrast, the momentum-based strategy ($b \leftarrow \tau b + (1-\tau)\bar{r}$) temporally smooths reward estimates, thereby attenuating variance and promoting more stable learning dynamics.

### 5 CONCLUSION

This paper proposes the first latent-space policy-gradient RL framework for semantic segmentation, where a stochastic policy network modulates intermediate feature representations according to task-aligned rewards. The overall network is optimized with a hybrid loss that combines policy-gradient and supervised segmentation objectives. By operating in latent space and proposing the hybrid loss, our method addresses key challenges that emerge when applying RL to segmentation tasks including action space complexity, pre-trained disparity and reward sparsity. Extensive experiments conducted across supervised, continual, and cross-domain segmentation settings obtain consistent performance improvements, indicating that the key benefits of policy-gradient RL validated in LLMs and MLLMs can be effectively transferred to conventional vision tasks through latent-space integration.

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

# A  SUPPLEMENTARY ABLATION STUDIES

Table 6: Ablation study on objective function configurations, where left denotes impact of individual loss components on the overall training objective, and right conducts ablation study on the weighting factors $\lambda_{\text{Policy}}$ and $\lambda_{\text{KL}}$, illustrating their influence on segmentation performance.

| Objective Function Configuration | mIoU (%) | $\lambda_{\text{Policy}}$ | $\lambda_{\text{KL}}$ | mIoU (%) |
|---|---|---|---|---|
| FCN | 67.8 | 0.0 | 0.0 | 76.0 |
| $\mathcal{L}_{\text{seg}}$ only where Eq. (14) $\rightarrow$ MultiheadAttention$(z_i, z_i, z_i)$ | $71.1 \pm 0.39$ | 0.1 | 0.1 | 77.8 |
| $\mathcal{L}_{\text{seg}} + \mathcal{L}_{\text{Policy}}$ | $77.9 \pm 0.55$ | 0.1 | 1.0 | 76.8 |
| $\mathcal{L}_{\text{seg}} + \mathcal{L}_{\text{Policy}} + \mathcal{L}_{\text{KL}}$ | $\mathbf{78.5 \pm 0.43}$ | 0.1 | 10.0 | 73.2 |
| – | – | 1.0 | 0.1 | 77.2 |
| – | – | 1.0 | 1.0 | **78.5** |
| – | – | 1.0 | 10.0 | 76.9 |
| – | – | 10.0 | 0.1 | 77.0 |
| – | – | 10.0 | 1.0 | 77.4 |
| – | – | 10.0 | 10.0 | 77.9 |
| – | – | 100.0 | 100.0 | 63.2 |
| – | – | 1000.0 | 1000.0 | 4.9 |

**Objective Function Configuration.** Table 6 demonstrates a series of ablation experiments designed to assess the individual contributions of each component in Eq. (6), which comprises three key terms, *i.e.*, the supervised segmentation loss $\mathcal{L}_{\text{seg}}$, the KL divergence regularization loss $\mathcal{L}_{\text{KL}}$, and the policy optimization loss $\mathcal{L}_{\text{Policy}}$. To elucidate the role of each component, we incrementally incorporate the components and report the corresponding mIoU performance on the PASCAL VOC benchmark.

As shown in Table 6, employing $\mathcal{L}_{\text{seg}}$ alone yields a baseline mIoU of 71.1%. Augmenting $\mathcal{L}_{\text{seg}}$ with $\mathcal{L}_{\text{Policy}}$ improves mIoU to 77.9%, showing that RL signals provide valuable guidance for modulating semantic features. The full formulation which further includes $\mathcal{L}_{\text{KL}}$ to regularize deviations from an EMA-smoothed policy, brings the highest mIoU of 78.5%, underscoring the stabilizing effect of KL regularization in policy learning.

Moreover, we analyze the sensitivity of model performance to the weighting coefficients $\lambda_{\text{Policy}}$ and $\lambda_{\text{KL}}$. As observed in Table 6, the optimal performance emerges when both coefficients are set to 1.0, suggesting a well-calibrated trade-off between supervised learning and RL signals. In contrast, it is observed that assigning disproportionately large weights exemplified by 1000 to either $\lambda_{\text{Policy}}$ or $\lambda_{\text{KL}}$ leads to marked mIoU degradation even when the total gradient magnitude is carefully controlled to remain comparable to that of the default configuration. This result suggests that LLMs and MLLMs, owing to their large-scale pre-training, possess effective priors over token-level generation, which in turn provide well-initialized action distributions for RL to operate effectively, even in the absence of strong supervised guidance. In contrast, conventional segmentation networks are typically initialized from weaker visual backbones, resulting in poorly calibrated action spaces and overly dominant RL signals, despite maintaining comparable gradient magnitudes, will disrupt the optimization process, overpower supervised signals, hinder stable learning and meaningful policy exploration.

In conclusion, these ablation studies highlight the critical role of hybrid loss design in our framework and demonstrate that carefully balancing RL objectives with supervised signals is essential for stable training and effective policy optimization in dense prediction tasks.

Table 7: Ablation studies examining the impact of different modulation strategies for incorporating latent-space policy signals $s_i$ into the decoding process. We compare four strategies: Concatenation, Add, Multiplication and MultiheadAttention described in Eq. (14).

| Modulation Methodology | PASCAL VOC (mIoU) |
|---|---|
| FCN | 67.8 |
| Add ($z_i^{\text{mod}} = z_i + s_i$) | 68.9 |
| Multiplication ($z_i^{\text{mod}} = z_i \odot (1 + \tanh(s_i))$) | 71.8 |
| Concatenation ($z_i^{\text{mod}} = \text{Concat}(z_i, s_i)$) | 69.1 |
| MultiheadAttention ($z_i^{\text{mod}} = \text{MultiheadAttention}(z_i - s_i', z_i, z_i)$) | **78.5** |

**Modulation Mechanism.** To evaluate the effectiveness of our proposed attention-based modulation mechanism, we conduct an ablation study comparing four distinct strategies for integrating the latent policy signal $s_i$ into the feature representation $z_i$ during the decoding stage, including,

- **Add:** The modulation signal is reshaped to match $z_i$ and added element-wise,

$$z_i^{\text{mod}} = z_i + s_i, \tag{20}$$

- **Multiplication:** The modulation signal is employed as a spatially-varying scaling factor,

$$z_i^{\text{mod}} = z_i \odot (1 + \tanh(s_i)), \tag{21}$$

  where $\odot$ denotes element-wise multiplication.

- **Concatenation:** The raw $s_i$ is concatenated with $z_i$ along the channel dimension,

$$z_i^{\text{mod}} = \text{Concat}(z_i, s_i), \tag{22}$$

- **MultiheadAttention:** It is described in Eq. (14).

The ablation results presented in Table 7 indicate that all modulation strategies confer improvements over the baseline FCN. Among them, the attention-based modulation brings the highest 78.5% mean IoU, thereby substantially surpassing the alternative approaches. This underscores the superiority of context-aware modulation via attention mechanisms over simpler arithmetic-based modulation.

Table 8: Ablation study on the impact of constraining the log standard deviation $\log \sigma_i$ in Eq. (10).

| Scaling Mechanism for $\log \sigma_i$ | PASCAL VOC (mIoU) |
|---|---|
| FCN | 67.8 |
| Unbounded | $43.5 \pm 7.22$ |
| $\log \sigma_i = 4 \cdot \tanh(\text{fc}_{\log \sigma}(\overline{z}_i))$ (ours) | $\mathbf{78.5 \pm 0.43}$ |

**Design of Eq. (10).** To generate the latent-space modulation signal $s_i$, $\pi_\phi$ parameterizes a Gaussian distribution per channel by predicting the mean $\mu_i$ and log-variance $\log \sigma_i$ from the latent feature $z_i$. As shown in Eq. (10), a $\tanh$ activation followed by a scaling factor of 4 is applied to $\log \sigma_i$ thereby constraining it within a bounded range, which curbs excessive stochasticity in the sampled signal $s_i$, promoting stable and efficient policy learning.

To assess the impact of above constraint, we compare against an unconstrained variant that generates $\log \sigma_i$ directly. As shown in Table 8, the unbounded formulation results in unstable optimization and a marked performance drop, whereas our bounded variant achieves substantially higher accuracy.

This experiment underscores the critical role of variance regularization in balancing exploration and ensuring training stability in our latent-space RL framework.

