# OpenReview forum: "Latent-Space Reinforcement Learning for Image Segmentation"
_ICLR.cc/2026/Conference — Submitted to ICLR 2026_

### Official Review · Reviewer_988f · 2025-10-29

**Soundness:** 2
**Presentation:** 1
**Contribution:** 2
**Rating:** 2
**Confidence:** 3

**Summary:**

The paper proposes a latent-space reinforcement learning (RL) framework for semantic segmentation. The key idea is to integrate a stochastic policy network into the latent feature space of a segmentation model (e.g., FCN, SegFormer, SETR). The policy modulates intermediate representations using PPO-based optimization guided by segmentation-aligned rewards (IoU, Dice, CE, Boundary). The paper claims improvements in segmentation accuracy, continual learning stability, and cross-domain generalization.

**Strengths:**

1. Structured and readable presentation:
The paper is clearly written, with well-organized sections, consistent mathematical notation, and visually neat framework figures (Fig.1–2).

2. Stable optimization considerations:
The use of EMA policy updates, KL regularization, and clipped advantages demonstrates awareness of RL instability issues in vision.

**Weaknesses:**

1. Lack of qualitative visualization
Despite being a computer vision paper, the submission contains no qualitative results whatsoever—no segmentation mask comparisons, no failure cases, no visual analysis.
This omission makes it impossible to assess whether the proposed latent-space modulation leads to meaningful perceptual or structural improvements.
For segmentation, qualitative evidence is not optional but essential.
The absence of visual results severely undermines the credibility of all numerical claims.

2. Inconsistent experimental improvements
FCN shows large mIoU gains (+10%), while SegFormer and SETR see marginal improvements (<2%). This suggests the approach mainly helps weaker backbones and may not generalize to modern architectures. No analysis is given to explain this discrepancy.
3. Computation cost (GPU hours, training stability, variance) is not discussed, though RL training is notoriously expensive.

4. Limited novelty beyond engineering combination
The method largely repackages existing PPO-based RL mechanisms within a segmentation pipeline. No new algorithmic principle or theoretical contribution is provided. Thus, the work’s originality lies mainly in applying standard RL to a new setting, not in methodological innovation.

5. Although the proposed approach is evaluated on three representative segmentation architectures (FCN, SETR, SegFormer), the comparison breadth remains limited.
Including more competitive baselines such as DeepLabV3+, HRNet, or UperNet — or comparing against other RL-based or adaptive optimization approaches — would strengthen the empirical validation and generality claim.

**Questions:**

Can you provide qualitative segmentation visualizations comparing baseline vs. your method?
Without them, it is impossible to evaluate the perceptual quality of results.

How is the latent-space modulation different from standard attention or FiLM conditioning mechanisms?
Is the Gaussian sampling truly necessary?

Why not adopt a deterministic policy if each image forms a single-step decision (contextual bandit)?
What is the actual exploration benefit here?

What is the computational overhead of adding the RL module (in GPU hours or FLOPs)?
Is the 1–2% mIoU gain on strong models worth the added complexity?

Could the improvements simply come from additional regularization (KL + noise) rather than from actual RL optimization?

---

### Official Review · Reviewer_Xthc · 2025-11-01

**Soundness:** 3
**Presentation:** 3
**Contribution:** 2
**Rating:** 4
**Confidence:** 4

**Summary:**

This paper explores the adaptation of policy-gradient reinforcement learning which is commonly used for LLM and MLLM fine-tuning to image segmentation. The authors propose a latent-space reinforcement learning (LSRL) framework that integrates RL into segmentation networks by acting on intermediate feature representations rather than raw pixels. The framework is evaluated across supervised, continual, and cross-domain segmentation settings using multiple backbones (FCN, Segformer, SETR). Results show consistent gains and improved cross-domain generalization, suggesting that benefits of RL-based optimization can transfer from language models to vision.

**Strengths:**

- The idea of conducting RL optimization in latent space is intuitively appealing—it mitigates high-dimensional pixel-level action spaces and connects segmentation with the recent trend of RL-based fine-tuning in LLMs.

- The integration of PPO with EMA-stabilized reference policies and advantage clipping is technically sound.  The hybrid loss effectively combines supervised learning and RL signals, as shown in ablations. The design of multi-metric rewards and attention-based modulation are empirically justified.

- Evaluations cover three standard segmentation datasets (ADE20K, Cityscapes, PASCAL VOC) and multiple architectures.  Ablation studies are detailed and confirm the contribution of each component (reward terms, baseline, modulation mechanism, bounded variance, etc.).

- The paper is well-structured and self-contained, with clear formulations and sufficient background. Figures are informative, and quantitative results are systematically organized.

**Weaknesses:**

- Conceptual novelty is limited. The proposed method mainly combines standard components: latent feature modulation, PPO optimization, and hybrid supervised losses.  Meanwhile, “Latent-space RL” is presented as novel, but it’s essentially an application of policy-gradient RL with minor engineering modifications to fit segmentation. There’s no fundamentally new RL formulation or theoretical insight.

- The paper empirically shows improvements but lacks a mechanistic analysis. Why exactly does policy-gradient optimization improve dense prediction compared to gradient descent?  The argument that RL provides better “credit assignment” or “generalization” remains speculative without deeper evidence (e.g., feature-space visualization or variance reduction analysis).

- The reward is a weighted sum of four metrics (IoU, Dice, CE, boundary), but weights are set equally (λ=1) without tuning or rationale. It is unclear how variance in these composite rewards affects policy stability.

- The framework adds substantial overhead (policy network, PPO optimization, multi-term loss) for relatively modest improvements (~1–2% mIoU on stronger backbones). For simple baselines (e.g., FCN), the gains are larger, but it’s unclear whether they stem from under-optimized baselines or genuine RL benefit.

- The paper does not compare LSRL against simpler non-RL regularization approaches (e.g., entropy minimization, self-training, or differentiable reward shaping). Without such comparisons, it’s difficult to isolate the contribution of RL itself from the effects of additional supervision or stochastic modulation.

**Questions:**

## Questions:
- Could you provide *qualitative visualizations* showing how the latent policy modulates features or affects segmentation masks?
- How sensitive is the framework to the reward coefficients $\lambda_{IoU}$, $\lambda_{Dice}$, etc.?
- What is the additional training cost (in FLOPs, GPU hours) compared to standard supervised training?
- Does the method still help when trained on large-scale datasets like COCO-Stuff or ADE20K full?

## Suggestions:

- The paper frequently references reinforcement learning for LLMs, but this analogy is superficial: segmentation does not involve long-term dependencies or human preference signals. This rhetorical framing may overstate the conceptual connection.

---

### Official Review · Reviewer_h9Ps · 2025-11-09

**Soundness:** 2
**Presentation:** 2
**Contribution:** 2
**Rating:** 4
**Confidence:** 4

**Summary:**

This paper presents an interesting and timely exploration of policy-gradient reinforcement learning (RL) applied to the domain of semantic segmentation. The authors introduce a novel latent-space RL framework that operates directly on intermediate feature representations of segmentation networks rather than at the pixel level, which can be a clever approach to address the high-dimensional action space challenge inherent in dense prediction tasks. The integration of policy-gradient RL directly within conventional segmentation networks bypasses the need of large foundation models. Several experiments show improvements as compared to the baseline methods.

**Strengths:**

1. This paper introduces a novel latent-space RL framework that operates directly on intermediate feature representations of segmentation networks rather than at the pixel level.
2. This paper tries to integrate RL into conventional CV architectures such as FCN, which is not well explored but interesting, bypassing the need of foundation models. For example, VLM-R1 uses Qwen as backbone. Models such as FCN possesses much less parameters and can be applied for Robotics vision.
3. The proposed method shows significant improvement for FCN architecture across ADE20K, Cityscapes, PASCAL VOC. Improvements over other frameworks such as SegFormer are also visible.

**Weaknesses:**

Major weakness:
1. The baseline for comparison is too weak. FCN (2015), SegFormer (2021), and SETR-Naive (2021) are currently not the state-of-the-art methods on the benchmarks. Moreover, the experiments on the three frameworks show more and more marginal improvements as the performance of baseline ascends.
2. The reported performance gains likely stem primarily from the multiple task-aligned rewards (especially $r^{IoU}$ which directly optimizes the evaluation metric) rather than the RL framework itself. These rewards could be straightforwardly converted into standard DL loss functions (e.g., 1-IoU loss with straight-through estimators or Dice loss), eliminating the need for complex RL machinery. The paper fails to adequately address this fundamental concern or provide ablation studies isolating the contribution of the RL framework versus the reward engineering.
3. The paper lacks many visualizations. We can not judge where the performance gain comes from. Is it better with boundaries? Or it locate the semantic instance in a more precise manner?

Minor weakness:
1. To fully reproduce this work, we may need similar resources. Two NVIDIA H200 can make it hard for ordinary groups to verify and reproduce.
2. To improve stability, the author introduces a learnable gate $\alpha$ to control the RL-induced results. I wonder the eventual value of $\alpha$.  After all, when $\alpha=0$, the model will be the same as the baseline. Also, what would be the effect when $\alpha$ is fixed?
3. The authors claims to treat the process as a one-step Markov Decision Process, and claim that the decoding process is the transition (line 216-217). However, as far as I know, transition should be defined as transition between states depending on the action and previous state. There is no "states" after decoding as the prediction has been made after decoding. I'm not sure if this is a typo or a misunderstanding.

**Questions:**

1. How does the proposed method work on current state-of-the-art methods, such as VLTSeg? Is there any visible improvements?
2. How does the baseline model perform when using other losses or multiple losses that align with the RL rewards? See major weakness.
3. See minor weakness 2.
4. See major weakness 3. I would be glad if I could see more visualization results.

---

### Meta-Review · Area_Chair_7kZs · 2026-01-07

**Summary:**

The authors did not submit the rebuttal. All three reviewers were unconvinced on the positive side; they agreed that this work requires additional effort to meet the acceptance bar of ICLR. Thus, I am inclined not to accept this draft at this stage. Thank you for your effort. It is an interesting work. I hope the input from the reviewers will help you further improve this work.

**Reviewer Concerns:**

This work has limited novelty and experimental results.

**Reviewer Scores:**

The reviewers' score reflects the limitations of this work.

---

### Decision · Program_Chairs · 2026-01-26

Reject